# *Staphylococcus epidermidis* Cicaria, a Novel Strain Derived from the Human Microbiome, and Its Efficacy as a Treatment for Hair Loss

**DOI:** 10.3390/molecules27165136

**Published:** 2022-08-12

**Authors:** HyungWoo Jo, Seon Yu Kim, Byung Ha Kang, Chaeyun Baek, Jeong Eun Kwon, Jin Woo Jeang, Young Mok Heo, Hye-Been Kim, Chan Yeong Heo, So Min Kang, Byung Ho Shin, Da Yeong Nam, Yeong-Geun Lee, Se Chan Kang, Dong-Geol Lee

**Affiliations:** 1COSMAX BTI, R&I Center, Seongnam 13486, Gyeonggi, Korea; 2Department of Oriental Medicine Biotechnology, College of Life Science, Kyung Hee University, Yongin 17104, Gyeonggi, Korea; 3COSMAX, Seongnam 13486, Pangyo-ro, Korea; 4H&BIO Corporation/R&D Center, Seongnam 13605, Gyeonggi, Korea; 5Department of Plastic and Reconstructive Surgery, Seoul National University Bundang Hospital, Seongnam 13620, Gyeonggi, Korea; 6Korean Skin Research Center (KSRC), Seongnam 13558, Gyeonggi, Korea

**Keywords:** adenosine, biotin, cicaria, hair microbes, microbiome, skin-microbiome, *Staphylococcus*

## Abstract

The skin tissue of the scalp is unique from other skin tissues because it coexists with hair, and many differences in microbial composition have been confirmed. In scalp tissues, hair loss occurs due to a combination of internal and external factors, and several studies are being conducted to counteract this. However, not many studies have addressed hair loss from the perspective of the microbiome. In this study, subjects with hair loss and those with normal scalps were set as experimental and control groups, respectively. In the experimental group, hair loss had progressed, and there was a large difference in microbiome composition compared to the group with normal scalps. In particular, differences in *Accumulibacter*, *Staphylococcus*, and *Corynebacterium* were found. From *Staphylococcus epidermidis* Cicaria, two active components were isolated as a result of repeated column chromatography. Spectroscopic data led to the determination of chemical structures for adenosine and biotin. Fractions were obtained, and ex vivo tests were conducted using hair follicles derived from human scalp tissue. When the microbiome adenosine-treated group was compared to the control group, hair follicle length was increased, and hair root diameter was maintained during the experimental periods. In addition, the Cicaria culture medium and the microbial adenosine- and biotin-treated groups maintained the anagen phase, reducing progression to the catagen phase in the hair growth cycle. In conclusion, it was confirmed that the Cicaria culture medium and the microbial adenosine and biotin derived from the culture were effective in inhibiting hair loss.

## 1. Introduction

Hair loss refers to a phenomenon in which hair falls out abnormally because of degraded function of a germinal matrix cell, caused by receiving nutrients from a dermal follicle papilla cell, which prevents the hair from growing properly and it falls out. Hair loss can have a psychologically adverse effect on the social activities of an individual, beyond simple cosmetic problems [1]. Hair loss occurs mainly due to genetic factors and aging, but the number of hair loss cases has increased among young people in their 20s and 30s due to environmental and psychological factors such as fine dust and stress. The use of drug therapy with Minoxidil and Finasteride (MSD) is currently being attempted to treat, alleviate, or prevent hair loss. Minoxidil is assumed to induce hair growth through vasodilation and potassium channel opening, and Finasteride inhibits the activity of 5α-reductase, an enzyme acting on male hormone metabolism, to prevent male-type hair loss [2]. However, both have crucial disadvantages. Minoxidil is difficult to use continuously and stably due to side effects such as dryness, dandruff, scalp irritation, and itching, and Propecia has side effects such as erectile dysfunction, decreased sexual desire, gynecosis, and low ejaculate volume [3].

Human skin is home to millions of bacteria, fungi, and viruses that make up the skin microbiome. Similar to intestinal microbes, skin microbes play an essential role in protecting against pathogen invasion, both in the human immune system and in the decomposition of natural products [4,5,6]. As the largest organ in the human body, skin serves as a physical barrier to beneficial microbes and pathogen invasion. If this physical barrier collapses or the balance between symbiosis and pathogens is disrupted, skin diseases or general disorders can occur [7,8].

Among them, the scalp is a skin tissue that coexists with hair among skin tissues, and many microbial composition differences from other skin tissues are confirmed in our pilot studies. In addition, scalp conditions such as decreased physiological function, local blood flow disorder caused by scalp tension, and poor scalp nutrition are known representative causes of hair loss [9,10,11,12]. Nevertheless, not many studies have been conducted on the inhibition of hair loss from a microbiome perspective. Thus in this study, focusing on the scalp as the skin of the head where hair grows, this study hypothesized that the scalp is affected by the microbiome, and that this is associated with hair loss [13]. *Cutibacterium* spp. and *Staphylococcus* spp. account for about 90% of the microbiome in the healthy scalp, in addition to *Corynebacterium* spp., *Streptococcus* spp., *Acinetobacter* spp., and *Prevotella* spp. [13]. The only research on the scalp microbiome has involved studies on simple compositional changes related to scalp inflammation or disease.

Therefore, this study attempted to investigate the effect of the microbiome on the scalp and hair through metabolic substance analysis and mechanical research using scalp cells.

## 2. Results and Discussion

### 2.1. Scalp Microbiome Analysis of Subjects with and without Hair Loss

A total of 28,000 sequencing reads were obtained for 40 scalp samples for ordinary and hair loss patients, confirming secure data for analysis (Appendix A). There was no difference in species diversity or species abundance between the two groups of samples, but there was a difference between the microbial clusters constituting the microbiome.

The microbiome of the control and hair loss group samples consisted of 16 genera that mostly belonged to the four phyla of Proteobacteria, Firmicutes, Bacteroides, and Actinobacteria (Appendix A).

Between the two groups, there was a greater than 10% difference in the distributions of Proteobacteria and Actinobacteria phyla. The control group had an abundance of Proteobacteria, and the hair loss group had an abundance of Actinobacteria (Appendix A).

As shown in Appendix A, the six classes (Actinobacteria, Gammaproteobacteria, Clostridia, Betaproteobacteria, Bacilli, and Alphaproteobacteria) were showing dominant tendency at the class level (Appendix A). There was a difference in the distribution of the Actinobacteria, Betaproteobacteria, Bacilli, and Alphaproteobacteria classes between the two groups. There were many Betaproteobacteria and Alphaproteobacteria in the control group, whereas there were more Actinobacteria in the hair loss group (Appendix A). Eleven genera dominated in common: *Rugosibacter, Cutibacterium, Sphingomonas, Accumulibacter, Curvibacter, Staphylococcus, Bradyrhizobium, Ralstonia, Afipia, Brevundimonas,* and *Enhydrobacter* (Appendix A). *Cutibacterium, Staphylococcus*, and *Brevundimonas* were more common in the hair loss group, and *Streptococcus* and *Rothia* were present only in the hair loss group. *Rugosibacter* was most common in the control group. At the species level, *Rugosibacter aromaticivorans, Cutibacterium acnes, Sphingomonas pruni, Staphylococcus aureus, Bradyrhizobium japonicum, Accumulibacter phosphatis*, Afipia PAC001985_s, Acumulibacter_uc, *Ralstonia solanacearum, Brevundimonas vesicularis*, FJ660572_g FJ660572_s, and *Enhydrobacter aerosaccus* were commonly dominant (Appendix A).

### 2.2. Selection of Target Strains of Hair Loss Suppression

#### 2.2.1. Separation and Identification of Bacterial Strains

A total of 300 strains were isolated from the scalps of the subjects. Among these, Staphylococcus accounted for 47.9%. This is different from the results of the scalp analysis, which identified Staphylococcus in about 4% of the microbiome in the normal and hair loss group. In addition to Staphylococcus, 30 other genera were identified (Figure 1).

#### 2.2.2. Isolated Bacterial Strains That Promote Epidermal Cell Growth Factor

We assessed whether the culture supernatants of the 300 strains obtained from the microbiome showed efficacy on dermal papilla cells by confirming the mRNA expression of epidermal cell growth factors. A Staphylococcus epidermidis strain with the highest probability was selected and named Cicaria. It showed similar efficacy to Minoxidil, which was used as a positive control (data not shown).

As shown in Figure 2A, at 24 h, cell proliferation tended to increase as the concentration increased, while cell growth tended to be inhibited at 72 h (Figure 2A). Also, it was confirmed that, as the concentration increased, the expression of the vasodilator gene also increased (Figure 2B).

To confirm the expression of the epidermal cell growth factor gene, the Cicaria culture supernatant was used to treat human papilla cells in conditions identical to those above. As concentrations increased to 0.1–1%, epidermal cell growth factor gene expression also increased; at a concentration of 10%, expression decreased (Figure 2C). The Cicaria culture supernatant was confirmed to be effective in the hair tissue because it induced the proliferation of human papilla cells, increased the expression of the vasodilator gene, and increased the expression of the epidermal growth factor gene. Eye and skin irritation tests were performed before conducting clinical trials using Cicaria culture supernatant. It was confirmed that there was no eye or skin irritation because both tests were negative (Appendix A). Minoxidil had fatal adverse effects such as dryness, dandruff, scalp irritation, and itching despite its high efficacy. Therefore, a Cicaria culture medium which has no toxicity would be a good alternative.

### 2.3. Microbiome Analysis of before/after Clinical Trial of Shamppo Containing of Cicaria supernant

Through multivariate statistical analysis, it was confirmed that the distribution of the microbiome itself moved when a clinical sample containing Cicaria culture supernatant was used for 4 weeks. With this, it was expected that Cicaria could have a positive effect on the changes in the microbiome of the scalp (Figure 3).

In addition, the Shannon, Chao1, and Simpson indices were evaluated to confirm the diversity, equality, and abundance of bacteria. The Shannon index tended to increase from 2.6 before using shampoo containing Cicaria strain culture supernatant to 3.25 after use, indicating that species diversity increased. The Chao1 index also confirmed that the shampoo containing the Cicaria strain culture supernatant increased from 560 before use to 620 after use. It was confirmed that the Simpson index decreased from 0.19 to 0.11 before using shampoo containing the Cicaria strain culture supernatant. Via comparison with other indices, since the species became diverse, it was confirmed that the distribution of the species became equal without the dominant species (Figure 4).

### 2.4. Identification and Analysis of Active Components in Cicaria

After lyophilization, 36 g of lyophilized supernatant was obtained from 20 L of Cicaria culture supernatant. It was confirmed that p-dimethylaminobenzaldehyde (Ehrlich’s reagent) has the most notable d-biotin aspect to be identified. Using Ehrlich’s reagent, thin layer chromatography (TLC) analysis was performed on fractions of lyophilized Cicaria supernatant.

Concentrated Staphylococcus epidermidis Cicaria supernatant was divided into n-hexane and H_2_O fractions. Repeated SiO_2_, octadecyl SiO_2_ (ODS), Sephadex LH-20, and Diaion HP-20 column chromatographies (c.c.) on each fraction gave the isolation of two active components (Figure 5). Through ^1^H nuclear magnetic resistance (NMR), ^13^C NMR, ^1^H-^1^H gCOSY, gHSQC, and gHMBC NMR analyses, it was confirmed that the active compounds were adenosine and biotin. 

**Adenosine:** White amorphous powder; ^1^H-NMR (600 MHz, DMSO-*d*_6_, δ_H_) 8.34 (1H, s, H-8), 8.14 (1H, s, H-8), 7.35 (1H, brs, H-NH_2_), 5.88 (1H, d, J = 6.6 Hz, H-rib-1), 5.47 (3H, overlapped, H-rib-2,3,5), 5.19 (1H, brd, J = 4.2 Hz, H-rib-3), 4.61 (1H, ddd, J = 7.2, 3.6 Hz, H-rib-2), 3.97 (1H, brdd, J = 7.2, 3.6 Hz), 3.67 (1H, ddd, J = 12.0, 3.6, 3.6 Hz, H-rib-5a), 3.56 (1H, ddd, J = 12.0, 7.2, 3.6 Hz, H-rib-5b); ^13^C-NMR (150 MHz, DMSO-d_6_, δ_C_) 158.2 (C-6), 154.4 (C-2), 151.0 (C-4), 141.9 (C-8), 121.3 (C-5), 89.9 (C-rib-1), 87.9 (C-rib-4), 75.4 (C-rib-4), 72.7 (C-rib-2), 63.7 (C-rib-5).

**D-biotin:** White amorphous powder; ^1^H-NMR (600 MHz, DMSO-*d*_6_, δ_H_) 4.61 (1H, brdd, J = 7.8, 4.8 Hz, H-4), 4.44 (1H, dd, J = 7.8, 4.2 Hz, H-3), 3.35 (1H, dt, J = 9.0, 4.2 Hz, H-2), 3.00 (1H, dd, J = 13.2, 4.8 Hz, H-5a), 2.75 (1H, overlapped, H-5b), 2.19 (2H, t, J = 7.4 Hz, H-a), 1.72 (1H, brdt, J = 12.6, 9.0 Hz, H-da), 1.60 (3H, overlapped, H-ba, bb, db), 1.43 (2H, brt, J = 8.0 Hz, H-c; ^13^C-NMR (150 MHz, DMSO-d_6_, δ_C_) 174.3 (C-10), 162.5 (C-2’), 61.2 (C-3), 59.2 (C-4), 55.6 (C-2), 39.9 (C-5), 33.6 (C-a), 28.2 (C-d), 28.2 (C-c), 24.6 (C-b).

### 2.5. Evaluation of the Efficacy of Fractions

#### 2.5.1. Analysis of Hair Follicle Length

The hair follicle length of the control group was analyzed and was found to be significantly increased by 23.71 and 22.35% by days 9 and 12 of hair follicle culture, respectively, compared to day 0. Hair follicle length in the microbiome adenosine 5 ppm group significantly increased by 26.91 and 26.84% by days 9 and day 12, respectively, compared to day 0. When groups were compared, the microbiome adenosine 5 ppm group showed a tendency to increase by 3.20 and 4.49% by days 9 and 12, respectively, compared to the control group (Appendix A).

#### 2.5.2. Analysis of Hair Root Diameter

The hair root diameter of the control group was analyzed and found to be significantly decreased by 8.19, 36.38 and 38.81% by days 3, 9, and 12 of hair follicle culture, respectively. The hair root diameter in the microbiome adenosine 5 ppm group significantly decreased by 1.79, 36.77 and 35.07% by days 3, 9, and 12, respectively. After 3, 9, and 12 days of hair follicle culture the hair root diameter in the microbiome adenosine 25 ppm group decreased significantly by 0.74, 15.58 and 18.33%, respectively, and the hair root diameter in the microbiome biotin 0.5 ppm group decreased significantly by 1.71, 27.81 and 33.87%, respectively. When the groups were compared, the microbiome adenosine 5 ppm, 25 ppm, and microbiome biotin 0.5 ppm groups showed significantly smaller differences in the rate that the hair root diameter changed compared to the control group (Figure 6).

#### 2.5.3. Analysis of Hair Follicle Growth Cycle Analysis

In hair follicle growth cycle analysis, the proportion of hair follicles that progressed to the catagen stage was 66.67, 66.67 and 77.78% by days 6, 9, and 12, respectively, and in the 25 ppm microbiome adenosine group it was 44.45, 55.56 and 55.56%, respectively, on the same days. When the groups were compared, in the 25 ppm microbiome adenosine group a lower proportion of 22.22, 11.11 and 22.22% of the hair follicles progressed to the catagen stage at the days 6, 9, and 12, respectively, compared to the control group (Table 1).

In the ex vivo assay, there was a limit to the test period because the growth cycle progressed with telogen after 12 days. As a result of culturing for up to 12 days, the hair follicle anagen cycle was maintained in the microbiome-treated group and increased (maintained) the hair root diameter. Therefore, if additional cultivation was possible, the final length can be expected to be longer.

Also, it was confirmed that the effectiveness of adenosine, isolated and purified from Cicaria, was superior to or equivalent to that of adenosine, which is commercially available. In addition, it was confirmed that the group treated with the microbiome culture medium showed almost similar efficacy to Minoxidil, which may infer the synergistic effect of adenosine and biotin in the culture medium. Such positive results can bring about the omission of the separation and purification process when applied to industry, and the shortening of this process is expected to bring cost advantages.

## 3. Materials and Methods

### 3.1. Recruitment of Subjects for Scalp Microbiome Analysis

The clinical trials were conducted by the Global Medical Research Center (GMRC), Seoul, Korea. Twenty-one individuals without hair loss and 21 with hair loss were included in the study in accordance with the guidelines for the recruitment of subjects and the IRB deliberation code (GIRB-19O01-AK). For the analysis of the subject’s scalp microbiome, the scalp was washed using 500 mL of sterile distilled water, and the washed solution was recovered and frozen for storage. Sampling was conducted in a controlled room at 20–24 °C and 45–55% RH. During the 30 min before sampling, the participants stayed still without moving their scalp. The scalp was washed using 500 mL of sterile distilled water for 3 min. The washed solution was stored at −80 °C before DNA extraction.

### 3.2. Microbiome Analysis

To recover the entire microbiome from the scalp-washed solution, only the organisms attached to the filter paper were recovered by filtering using a sterile 500 mL bottle-top vacuum filter with a 0.45 µm pore 33.2 cm² CA membrane that fits 45 mm diameter necks (Coning) and secured in a 50 mL conical tube. To secure the bacterial strains on the filter paper, 30 mL of sterile distilled water was added, and the strains were dispersed by shaking for 20 min using a vortex. The filter paper was removed, and the solution was centrifuged at 10,000× *g* for 15 min at 4 °C to ensure that the bacteria and the supernatant was removed. DNA was extracted from the bacteria according to the manual provided with the QIAamp PowerFecal Pro DNA Kit, and the DNA concentration was measured using Nanodrop. For microbiome analysis, a request was sent to Chunlab Co., Ltd. (Seoul, Korea) The extracted DNA was amplified with V3-V4 in the 16s rRNA gene region to produce a DNA library, and the library was purified using a Clean-Up Kit. Sequencing analysis was conducted using Miseq equipment. The raw sequencing data were analyzed using a microbiome analysis pipeline (Qiime2) provided by Chunlab Co., Ltd.

### 3.3. Separation and Identification of Strains Involved in Hair Loss Suppression

To obtain strains that can help to suppress hair loss, a sample of the scalp-wash solution was diluted serially and inoculated onto the tryptic soy agar, R2A medium. The inoculated medium was cultured at 37 °C for 5 days and then selectively separated into 300 different species in consideration of the morphological similarity of colonies. The 16S rRNA gene sequences of the isolated strains were referred to Macrogen Co., Ltd. (Seoul, Korea) and analyzed through Applied Biosystems 3730xl DNA Analyzer devices by cycle sequencing using the BigDye Terminator. To draw the phylogenetic tree of isolated strains, the sequence of the 16S rRNA genes of various species close to the isolated strains was investigated using sequence data registered in GenBank, and their sequences were aligned using the Bioedit and Club X programs (MEGA program version 11, Raynham MA, USA). The Kimura two-parameter model was used to track the evolutionary process of strains, and the systematic taxonomic position was determined by the neighbor-joining and maximum parsimony methods in the MEGA program.

### 3.4. Morphological Analysis of Cicaria

The Cicaria strain was grown on R2A agar for 2 days at 30 °C. The cells were pre-fixed using 4% glutaraldehyde solution and post-fixed using osmium tetroxide. Then they were washed using sterilized water and dehydrated with serial dilutions of ethanol (50, 60, 70, 80, 90 and 100%). Cells were dried to a critical point in CO_2_ and coated with gold in a sputter coater (SC502, Polaron) and were observed using a transmission electron microscope (LIBRA 120; Zeiss). The Cicaria strain was photographed using an SEM device to confirm the cell shape in the form of a coccus (Figure 7).

### 3.5. Assessment of Proliferation of Human Follicle Derma Papilla Cells

Human follicle derma papilla cells (HFDPC) were dispensed into 96 well plates by using 2 × 10^4^ cells/well, and then cultured for 24 h in an incubator at 37 °C and 5% CO_2_. After removing the medium containing nothing, the culture solution of the strain was added and further cultured for 24 h. A positive control group was set by adding 10 ppm of Minoxidil (6-piperidin-1-yl pyrimidine-2, 4-diamine 3-oxide, known as a hair development promoter) instead of the strain culture solution. After 24 and 72 h, the medium of each well was removed, a washing process using Dulbecco’s phosphate-buffered saline (DPBS) was performed once, and then medium containing 0.5 mg/mL MTT (Sigma-Aldrich, St. Louis, MO, USA) was added per well at 37 °C. Thereafter, the absorbance was measured at 570 nm using a spectrophotometer (Victor 3, Perkin-Elmer). The survival rate of cells was shown as a percentage of the average absorbance value and used to confirm the ability of HFDPC to proliferate. The Cicaria culture supernatant was used at concentrations of 0.1, 1 and 10% for 24 h and 72 h to treat human follicle derma papilla cells to confirm cell proliferation. Also, to evaluate the expression of the vasodepressor gene, different concentrations (0.1, 1 and 10%) of the Cicaria culture supernatant were used to treat human papilla cells for 24 h.

### 3.6. Evaluation of Growth Factor (VEGF, FGF7) Expression in Human Papilla Cells

HFDPC were divided into 6-well plates (3 × 10^5^ cells/well), and then cultured for 24 h under cell culture conditions. After 24 h, the medium was discarded, washed with phosphate-buffered saline (PBS), and then the cells were starved using a medium without fetal bovine serum (FBS). The next day, the culture solution was replaced with a medium containing the strain, and then further cultured for 24 h. A group of HFDPC containing 10 ppm Minoxidil was set as a positive control group. Subsequently, RNA was isolated from cells using an RNeasy Mini Kit (Qiagen, Germany), RNA was quantified at 260 nm using nanodrop, and cDNA was synthesized (C1000 Thermal Cyclers, Bio-Rad, Hercules, CA, USA) using 2 μg of RNA, respectively. A real-time polymerase chain reaction was performed in a real-time PCR machine using a mixture of VEGF or FGF7 primers and SYBR Green (Applied Biosystems, Foster City, CA, USA), a cyanine dye used to synthesize cDNA. The level of VEGF and FGF7 gene expression was evaluated. The sequences of the primers used are shown in Table 1. The levels of gene expression were quantified relative to the β-actin gene (Table 2).

### 3.7. Clinical Trial of Cicaria Supernant

To compare before and after clinical product use of Cicaria, a shampoo in the form of wash off was produced as below (Table 3).

The Cicaria culture supernatant which was main the ingredient of the shampoo was preservative using 1,2-hexanediol, and it was made of water type essence containing more than 93% of the culture solution, and the final material was named CICARIA-W (Table 4). The final raw material of CICARIA-W is a slightly yellowish liquid type raw material, based on the specificity of the culture, and the pH is within the range of 6.0 to 9.0, and the microbial detection test is confirmed to be within the range of 100 CFU/g or less in accordance with the cosmetics test guidelines. It was confirmed that residual pesticides such as heavy metals also entered the acceptance level of the cosmetics test guidelines.

Clinical trials were conducted at the Global Medical Research Center, and 21 people with hair loss and 21 people without were studied in accordance with the guidelines for securing IRB deliberation codes (GMRC: GIRB-19O01-AK) and recruiting subjects. They were required to use clinical products once a day for 4 weeks.

For scalp microbiome analysis of subjects before/after use, the scalp was washed using 500 mL of sterile distilled water, and the washing solution was recovered and stored in a freezer. In order to recover the entire microbiome from the scalp wash solution, only the microbiome attached to the filter paper was recovered to 50 cubic meters using a 500 mL Bottle Top Vacuum Filter, 0.45 mm Pore 33.2 cm CA Membrane, Fit 45 mm Diameter Necks, and Sterile. In order to secure the strain in the filter paper to which the microbiome was attached, 30 mL of sterile distilled water was added, and then the strain was dispersed in the sterile distilled water by shaking for 20 min using a vortexer.

Thereafter, the filter paper was removed and centrifuged under conditions of 10,000× *g*, 4 °C, and 15 min using a centrifuge to secure only the fungus, and the supernatant was removed. A DNA extraction was used according to the manual of the QIAamp PowerFecal Pro DNA Kit using only the secured bacteria, and the DNA concentration was measured using Nanodrop.

### 3.8. Isolation of Active Components of Cicaria

To produce the supernatant of the Cicaria culture, the isolated and cultured Cicaria strain was inoculated in 200 mL of the R2A liquid medium and cultured for 48 h at 30 °C and 160 rpm. Using this as a seed, the culture medium was inoculated in 20 L of the R2A liquid medium and cultured for 48 h at 30 °C and 160 rpm. This was then centrifuged at 8000× *g* for 20 min at 4 °C, and the supernatant was recovered. The bacteria were removed using a 0.45 μm filter. The recovered supernatant was lyophilized at −70 °C and 30 mTorr until dried completely.

The targeted efficacy of biotin could not be confirmed with H_2_SO_4_, which is generally used for color reactions. Therefore, by referring to its structural specificity of having two first-class amines, *p*-anisaldehyde, KMnO4, ferric spray, bromocresol green, potassium permanganate, dimethylaminocinnamaldehyde, and *p*-dimethylaminobenzaldehyde (Ehrlich’s reagent) were used for the color reaction and appropriate conditions for the detection were established.

Thirty-six grams of the freeze-dried product of the Cicaria strain culture supernatant were extracted twice using 100 mL of methanol, and the extracted layer was extracted twice using 100 mL of n-hexane. The solvent, divided into the aqueous solution layer and the organic solvent layer, was concentrated under reduced pressure to obtain n-hexane and the aqueous fraction. The fraction was identified using Ehrlich’s reagent to confirm whether the segmented material exhibited a color similar to that of biotin.

The silica gel (SiO_2_) resin for column chromatography (c.c.) Kiesel 60 (Merck, Darmstadt, Germany) and ODS were used with Lichroprep RP-18 (40–60 μM, Merck) for TLC. The UV lamp used was Spectroline (ModelENF-240 C/F, Spectronics Corporation, Westbury, NY, USA). A 600 MHz FT-NMR spectrometer (Advance 600, Bruker, Germany) was used as the NMR spectrum. All reagents used in the experiment were from Sigma-Aldrich.

To subdivide the separated material, 100 mL of aqueous fraction obtained in the previous experiment was extracted twice by using 100% MeOH to obtain a MeOH, H_2_O (residue) fraction. Five different MeOH fractions were obtained by using a Diaion HP20 c.c (Φ 4.0 × 50.0 cm, H_2_O → MeOH 2 L). As a result of the TLC, SiO_2_ c.c (Φ 4.0 × 50.0 cm, CHCl_3_:MeOH:H_2_O = 10:3:1 → 6:4:1) was performed on the fifth fraction with a clear d-biotin layer, and six more samples (MeFr. 5-1~5-6) were additionally obtained by performing reduced pressure concentration. The H_2_O (residue) fraction was also processed by using Diaion HP-20 c.c (Φ 4.0 × 20.0 cm, H_2_O → MeOH, 2 L of each) to obtain aqueous and MeOH fr. TLC confirmed that the fractions of Cicaria showed a color similar to d-biotin. The MeOH Fr. 5-2 fraction showed relatively vivid color development. In addition, another compound found in this fraction with the same R_f_ value as d-biotin was confirmed by UV. To separate these two compounds, the TLC condition was changed to ODS acetone:MeOH:H_2_O = 1:1:3 to obtain a difference in the R_f_ values between the two compounds.

### 3.9. Evaluation of the Fraction Efficacy

HaCaT, a human keratinizing cell line, was purchased from the American Type Culture Collection (ATCC, Rockville, MD, USA) and cultured with 10% FBS and 1% antibiotic/antimicrobial agents. Cells were subcultured at intervals of 2 to 3 days. To confirm the barrier-enhancing effect in human keratinocytes, HaCaT cells were first adjusted to 4 × 10^5^ cells/well using Dulbecco’s Modified Eagle’s medium (DMEM) with 10% FBS added, inoculated on a 6-well plate, and cultured for 24 h. After incubation, the sample was treated with 10 ppm of the Cicaria fraction and incubated for 24 h. RNA was extracted from cultured cells to quantify RNA and then reacted at 42 °C for 55 min and 70 °C for 15 min to synthesize cDNA. RT-PCR used a PCR machine (Step One Plus, Applied Biosystems, Foster City, CA, USA) with cyber green SYBRGreen Supermix) and Filaggrin or ABCA12 gene primers or cDNA for 5 min to activate the polymerase at 94 °C for 30 s, and to polymerize for 40 cycles of 95 °C for 30 s, 54 °C for 1 min, and 72 °C for 1 min.

### 3.10. Culturing Human Papilla Cells and Analysis of Hair Growth Factors

To evaluate the effect of the Cicaria fraction on the expression of hair growth factors, a genetic analysis of the hair growth factors was conducted in HFDPCs. First, HFDPCs were purchased from PromoCell (Heidelberg, Germany) and cultured in a DPC growth medium containing the supplement and 100 U/mL of penicillin and 100 ug/mL of streptomycin. For gene expression analysis, HFDPCs were inoculated in a 6-well plate at a density of 3 × 10^5^, and then cultured for 24 h in an incubator at 37 °C and 5% CO_2_. After 24 h, the culture medium was discarded, washed with DPBS, and replaced with a medium not containing supplements or a medium containing 10 ppm Minoxidil, as a positive control, or Cicaria fraction 1 ppm, and further cultured for 24 h. Subsequently, RNA was isolated from cells using an RNeasy Mini Kit (Qiagen); RNA was quantified at 260 nm using nanodrop, and the cDNA was synthesized in amplifiers (C1000 Thermal Cyclers, Bio-Rad, USA) using 2 μg of RNA. The degree of VEGF, FGF7, and HGF gene expression was finally evaluated by performing a real-time polymerase chain reaction in a real-time PCR machine using a mixture of FGF7 and HGF primers and cyber green (SYBR Green Supermix) to synthesize cDNA. Primer sequences and reaction conditions are shown in Table 1, and the expression levels of genes were analyzed relative to the β-actin gene.

### 3.11. Human Fibroblast Culture and Wound Recovery Analysis

To evaluate the effect of Cicaria on wound recovery, the proportion of cells that had moved to the wound area and formed into a single layer of cells was evaluated. First, Hs68, a human fibroblast cell line, was cultured with 10% FBS and 1% penicillin added to DMEM. To analyze wound recovery through the movement of cell populations, cultured cells were inoculated in 96-well plates at a density of 2 × 10^4^ and then cultured for 24 h in an incubator under 37 °C and 5% CO_2_ conditions to create a 100% confluent state. Afterward, the attached cells were scratched using a cell scraper and then washed with PBS to remove the buoyant cells in the wound area. Then, DMEM medium with 10 ppm of the test substance and 1% penicillin was added and cultured for an additional 20 h. As a control group, DMEM containing 1% penicillin without the sample was used in cells. At intervals of 2 h for 20 h, the extent to which cells transferred to the arbitrarily wounded area was observed using Incucyte, a cell imaging analysis equipment.

### 3.12. Isolation and Cultivation of Hair Follicles

Scalp tissue of Korean adults aged 20 to 60 without skin disease was used for research with approval from the Korean Skin Research Center (KSRC) IRB, Seoul, Korea (IRB approval number: HBABN01-210728-BR-E0323-01) and Seoul National University Bundang Hospital IRB, Bundang, Korea (IRB approval number: B-2107-694-304). After washing the scalp tissue twice with PBS, it was placed in a sterilized petri dish. Based on the dermis, hair follicles were isolated under an anatomical microscope. The adipose tissue around the isolated hair follicle tissue was removed as much as possible and cut to a certain length based on the dermal papillar. Hair follicles were cultured in William’s E medium at 37 °C and 5% CO_2_ in an incubator with 2 mM L-glutamine, 10 µg/mL insulin, 10 ng/mL hydrocortisone, and 1% antibiotics. For hair follicle cultivation, hair follicles were cultured in a medium treated with each test substance (1 mL/24-well plate) for up to 12 days. An untreated group, a positive control group, an indicator material control group, and a test material treatment group were included. On days 0, 3, 6, 9, and 12 of the culture, hair follicle observation, hair follicle length, and hair root diameter were measured through photography (Figure 8).

### 3.13. Measurement of Hair Follicle Length and Hair Root Diameter

After treating hair follicles separated from the human scalp tissue with 5 ppm of microbiome adenosine, 25 ppm, 100 ppm, 0.5 ppm of microbiome biotin, and 5 ppm of microbiome culture, hair length was observed at 3, 6, 9, and 12 days. The hair follicles were photographed using an anatomical microscope (SZ51, OLYMPUS, Tokyo, Japan) and ToupLite software (ToupTek, Zhejiang, China) on 0, 3, 6, 9, and 12 days of culture. The hair follicle length and hair follicle diameter were measured using the Image J (The National Institutes of Health, Bethesda, MD, USA) program. Repeated experiments were performed independently three times.

### 3.14. Evaluation of Hair Follicle Growth Cycle

After treating hair follicles separated from the human scalp tissue with 5 ppm, 25 ppm, 100 ppm of adenosine derived from Cicaria, 0.5 ppm of microbiome Biotin, and 5 ppm of microbiome culture solution, hair growth cycles were observed at days 3, 6, 9, and 12. The hair follicles were photographed using anatomical microscopes (SZ51, OLYMPUS, Japan) and ToupLite software (ToupTek, China) on days 0, 3, 6, 9, and 12 of cultivation. The hair follicle structure was distinguished by its hair root from the anagen to catagen. Repeated experiments were performed independently three times.

### 3.15. Statistical Analysis

The statistical significance of all data was verified using SPSS Package Program version 20 (IBM, Armonk, NY, USA). The normality of the data was verified through the Shapiro-Wilk test and kurtosis and skewness. Analysis of Variance (repeated measure ANOVA) was used for comparison between before and after time points and between groups (*p* < 0.05). The rates of change in hair follicle length, hair root diameter, and hair follicle organ growth cycle were calculated as follows.
Rate of change in hair follicle length (%) = (Hair follicle length at each time point/Average hair follicle length at day 0) × 100—Average percentage of hair follicle length at day 0;Rate of change in hair root diameter (%) = (Hair root diameter at each time point/Average hair root diameter at day 0) × 100—Average percentage of hair root diameter at day 0;Rate of hair follicle growth cycle (%) = (Number of hairs in catagen/Total number of hairs) × 100.

## 4. Conclusions

In conclusion, as a result of analyzing the distribution of the microbiome between normal and hair loss, it was confirmed that there was no difference in the diversity of microbial species between the two groups in terms of abundance and equality. However, in the distribution of bacteria, it was confirmed that the pattern was significantly different, and the dominance of a specific individual was higher than normal. Also, among the microorganisms with different dominant values, Cicaria was confirmed to be effective in suppressing hair loss.

Considering the evaluation results of the culture solution and the interaction between microorganisms and the secretion of various and complex substances in the actual scalp environment, it was predicted that more effective efficacy could be expected. In fact, a single fractional material derived from microbial adenosine and D-biotin showed better efficacy than synthetic compounds. Therefore, it is confirmed that there are unknown mechanisms or other functional groups involved.

In this study, we first attempted to apply microbiome technology to the suppression of hair loss and although the possibility has been confirmed, it seems necessary to clarify the results from a detailed cell signaling and metabolomics perspective. Also, since there are many unknown substances and additional unidentified substances, further studies are needed to investigate their potential therapeutic effects against hair loss, revealing their mechanisms and synergistic effects with active compounds.

## Figures and Tables

**Figure 1 molecules-27-05136-f001:**
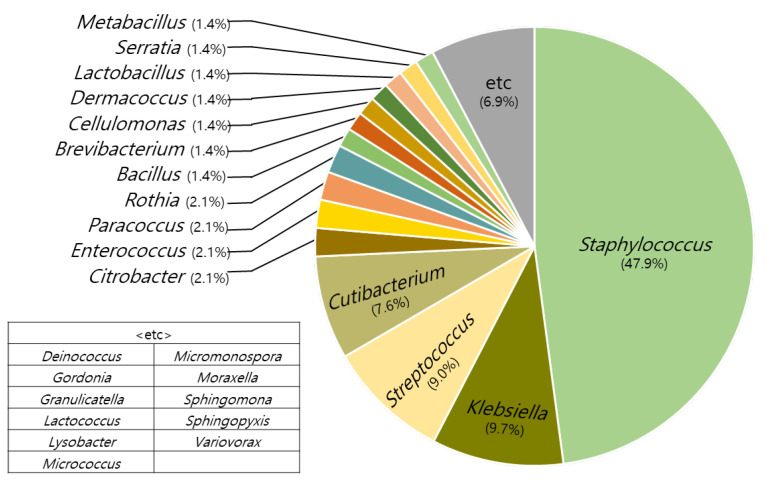
Genus types and distributions of isolated microbiomes.

**Figure 2 molecules-27-05136-f002:**
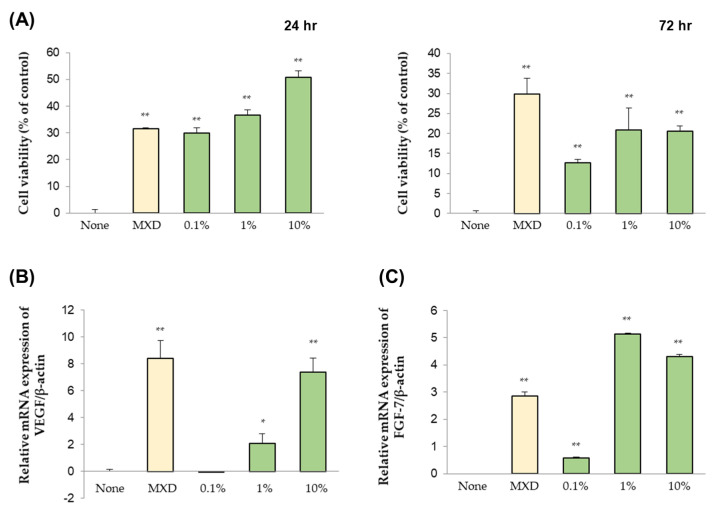
Cell proliferation efficacy of Cicaria culture supernatant by time (24 and 72 h) and concentration (**A**). Expression of the vasodilator gene (**B**) and epidermal cell growth factor gene (**C**) by the concentration of Cicaria culture supernatant. * *p* < 0.05, ** *p* < 0.01.

**Figure 3 molecules-27-05136-f003:**
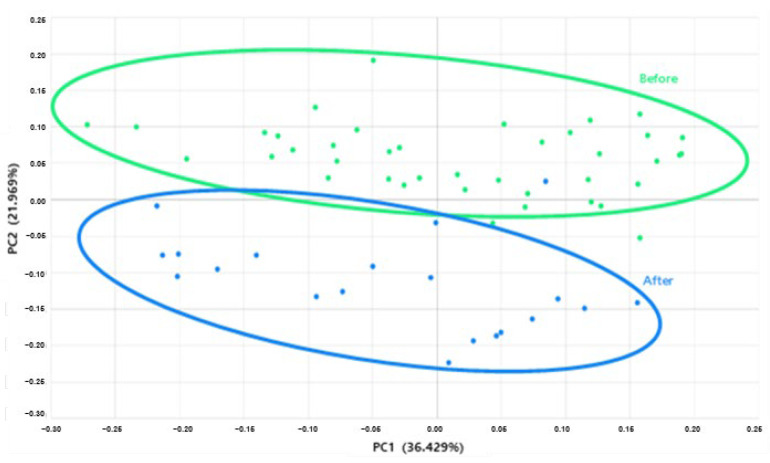
Multivariate statistical analysis of before/after clinical trial of shampoo containing of Cicaria supernant.

**Figure 4 molecules-27-05136-f004:**
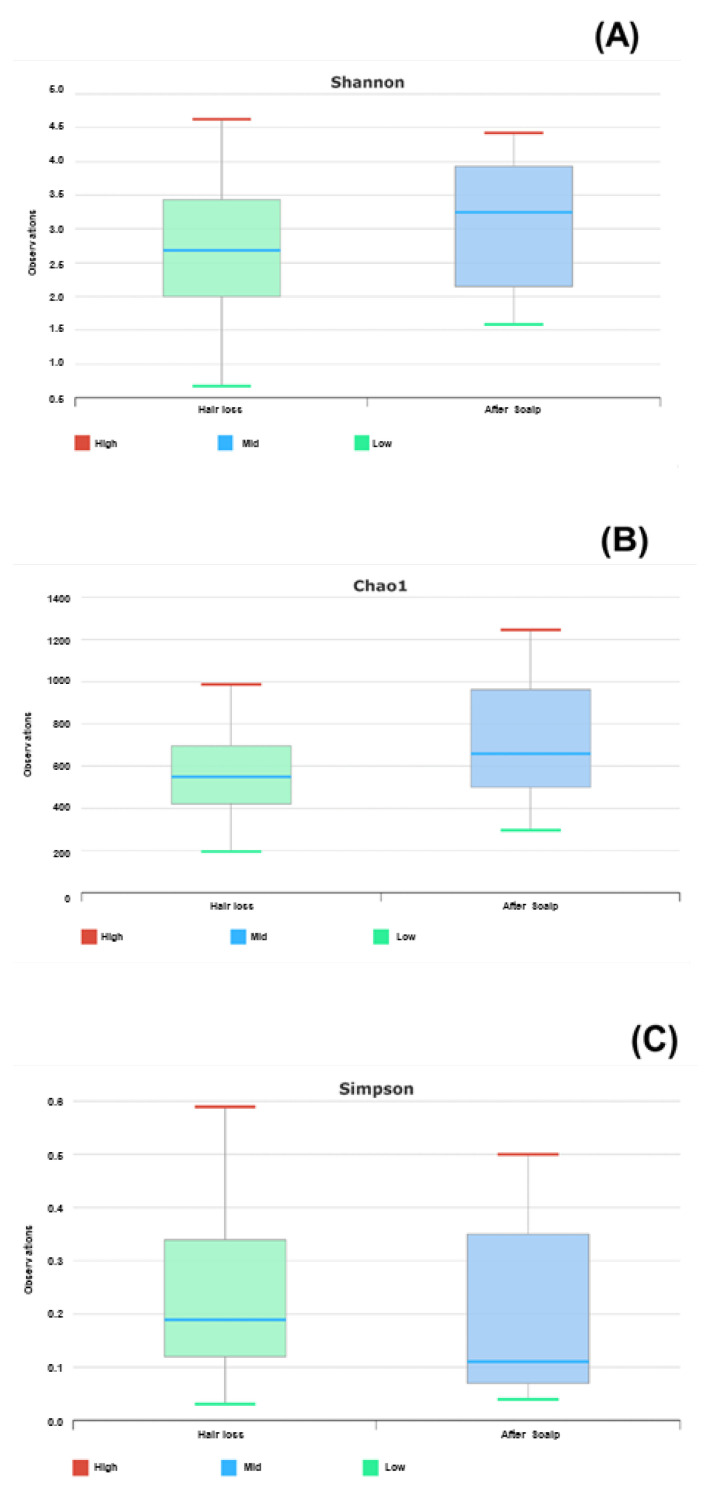
Species diversity (**A**), abundance (**B**), and uniformity (**C**) indices before/after the clinical trial of Cicaria supernatant shampoo clinical trial.

**Figure 5 molecules-27-05136-f005:**
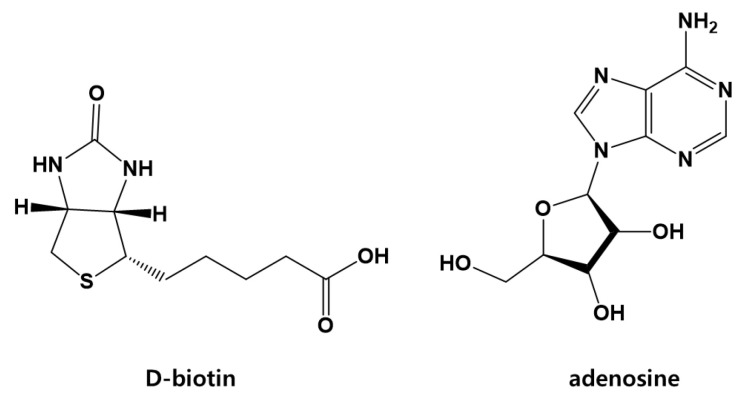
Chemical structures of the active components in S. epidermidis Cicaria.

**Figure 6 molecules-27-05136-f006:**
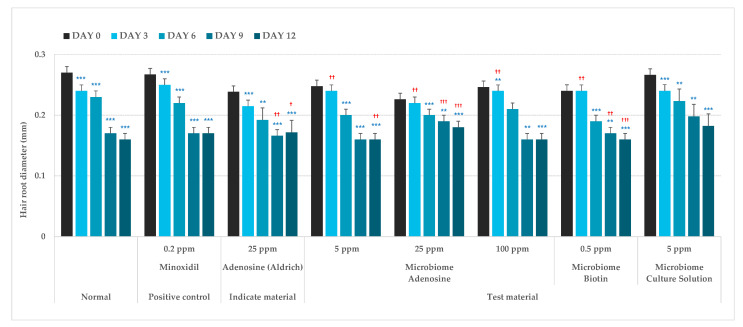
The assessment of hair root diameter (N = 9). ** *p* < 0.01, *** *p* < 0.001, ^†^
*p* < 0.05, ^††^
*p* < 0.01, ^†††^
*p* < 0.001.

**Figure 7 molecules-27-05136-f007:**
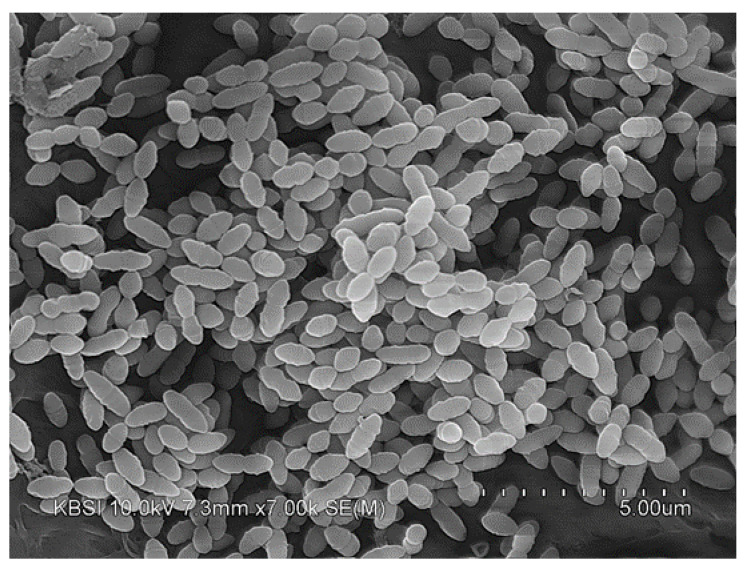
SEM photograph of the isolated Cicaria strain.

**Figure 8 molecules-27-05136-f008:**
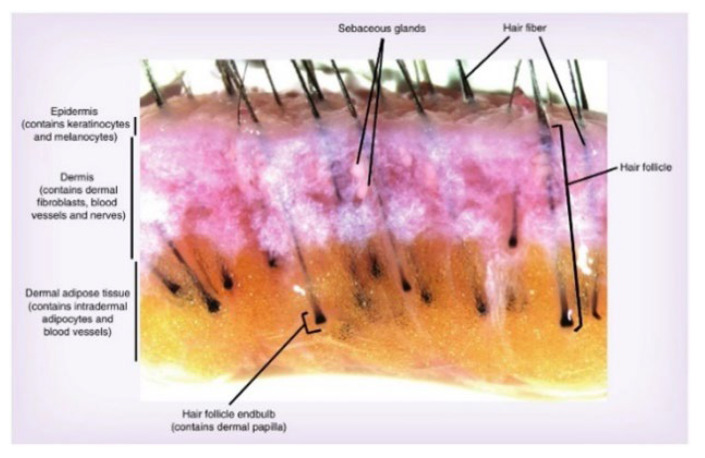
Cross-section of scalp tissue.

**Table 1 molecules-27-05136-t001:** The ratio of the number of follicle strands in the catagen phase to the hair follicle growth cycle at each time point.

Group	Test Material	Concentration	Day 3		Day 6		Day 9		Day 12	
			^1^ Strand	^2^ Rate (%)	Strand	Rate (%)	Strand	Rate (%)	Strand	Rate (%)
Normal	-	-	2/9	22.22	6/9	66.67	6/9	66.67	7/9	77.78
Positive control	Minoxidil	0.2 ppm	0/9	0.00	5/9	55.56	6/9	66.67	6/9	66.67
Indicate material	Adenosine (Aldrich)	25 ppm	0/9	0.00	3/9	33.33	7/9	77.78	8/9	88.89
Test material	Microbiome Adenosine	5 ppm	0/9	0.00	2/9	22.22	7/9	77.78	8/9	88.89
		25 ppm	2/9	22.22	4/9	44.45	5/9	55.56	5/9	55.56
		100 ppm	3/9	33.33	4/9	44.45	5/9	66.67	7/9	77.78
	Microbiome Biotin	0.5 ppm	0/9	0.00	6/9	66.67	6/9	66.67	7/9	77.78
	Microbiome Culture solution	5 ppm	0/9	0.00	5/9	55.56	5/9	55.56	6/9	66.67

^1^ Strand: A number of hairs in catagen stage/a total number of hairs. ^2^ Rate (%): A number of hairs in catagen stage/a total number of hairs × 100.

**Table 2 molecules-27-05136-t002:** Primer information for efficacy validation in vitro.

Gene Symbol	F/R	Sequence
VEGF	F	5′-GTGCCCACTGAGGAGTTCAAC-3′
	R	5′-CCCTATGTGCTGGCCTTGAT-3′
FGF7	F	5′-TCCTGCCAACTTTGCTCTACA-3′
	R	5′-CAGGGCTGGAACAGTTCACAT-3′
*β*-actin	F	5′-GGCCATCTCTTGCTCGAAGT-3′
	R	5′-GAGACCTTCAACACCCCAGC-3′

**Table 3 molecules-27-05136-t003:** Components of Cicaria shampoo.

No	Component	Ratio (*w*/*w*)
1	CICARIA-W	39.05
2	Polyquaternium-10	0.2
3	Sodium Citrate	0.1
4	Sodium Cocoyl Alaninate (solution)	40
5	Lauryl Hydroxysultaine (solution)	9
6	Coco-Betaine (solution)	9
7	Lauramide MIPA	0.8
8	Aspartic Acid	0.5
9	Caprylyl Glycol	0.3
10	Glyceryl Caprylate	0.3
11	Polyquaternium-22	0.5
12	Citric Acid	0.25

**Table 4 molecules-27-05136-t004:** Components of CICARIA-W.

Component	Contents (%)
*S. epidermidis* Cicaria supernant	93.8
Water	4.0
1,2-Hexandiol	2.0
Biotin	0.1
Arginine	0.1

## Data Availability

Not applicable.

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
