# Peer review of "Staphylococcus epidermidis Cicaria, a Novel Strain Derived from the Human Microbiome, and Its Efficacy as a Treatment for Hair Loss"

_molecules, 2022, doi:10.3390/molecules27165136_

Round 1
Reviewer 1 Report
In my review of article tilted 'Staphylococcus epidermidis Cicaria, a novel strain derived from the human microbiome, and its efficacy as a treatment for hair loss', I found the content and data of the ms interesting and worthy; however, there are huge flaws in the presentation of the ms. My comments are as follows:
Critical comments:
· Introduction: line 61 and 62, these line does not fit in the flow of context being presented. Shift these lines in the para where reasons of hair loss are presented
· More background studies should be reviewed not only stating the taxonomy of scalp microbiome but also the physiology of scalp microbes
· Line 122-123: Results should not describe M&M. Move it to M&M section
· Same is the case in line 126 to 127. This should be mentioned in M&M
· Line 145 to 151: Authors are just stating the M&Ms and not results
· The whole of the results and discussion section looks like authors are re-writing the M&Ms
· The whole results and discussion section should be re-written. There is no discussion of results at all
· The conclusion is just a repitition of results
Author Response
Thank you for your kind letter, with regard to our manuscript together with comments. We are thankful to you for the very valuable suggestions through the whole manuscript. Thank you again for your kind considerations.
We tried to revise the manuscript as much as possible in line with suggestion made by the reviewer. I am herewith enclosing improved manuscript.
- Reviewer 1
In my review of article titled 'Staphylococcus epidermidis Cicaria, a novel strain derived from the human microbiome, and its efficacy as a treatment for hair loss', I found the content and data of the ms interesting and worthy; however, there are huge flaws in the presentation of the ms. My comments are as follows:
Critical comments:
Introduction: line 61 and 62, these line does not fit in the flow of context being presented. Shift these lines in the para where reasons of hair loss are presented
→ Thank you for your comment. As indicated, I correct introduction part as below
Hair loss refers to a phenomenon in which hair falls out abnormally because of degraded function of a germinal matrix cell by receiving nutrients from a dermal follicle papilla cell, which prevents the hair from growing properly and it falls out. Hair loss can have a psychological adverse effect on the social activities of an individual beyond simple cosmetic problems [1]. Hair loss occurs mainly due to genetic factors and aging, but the number of hair loss cases has increased among young people in their 20s and 30s due to environmental and psychological factors such as fine dust and stress. The use of drug therapy with Minoxidil and Finasteride (MSD) is currently being attempted to treat, alleviate, or prevent hair loss. Minoxidil is assumed to induce hair growth through vasodilation and potassium channel opening, and Finasteride inhibits the activity of 5α-reductase, an enzyme acting on male hormone metabolism, to prevent male-type hair loss [2]. However, both have crucial disadvantages. Minoxidil is difficult to use continuously and stably due to side effects such as dryness, dandruff, scalp irritation, and itching, and Propecia has side effects such as erectile dysfunction, decreased sexual desire, gynecosis, and low ejaculate volume.
Human skin is home to millions of bacteria, fungi, and viruses that make up the skin microbiome Similar to intestinal microbes, skin microbes play an essential role in protecting against pathogen invasion, in the human immune system, and in decomposition of natural products [3–5]. As the largest organ in the human body, skin serves as a physical barrier to beneficial microbes and pathogen invasion. If this physical barrier collapses or the balance between symbiosis and pathogens is disrupted, skin diseases or general disorders can occur. In addition, scalp conditions such as decreased physiological function, local blood flow disorder caused by scalp tension, and poor scalp nutrition are known representative causes of hair loss [6].
In this study, focusing on the scalp as the skin of the head where hair grows, this study hypothesized that the scalp is affected by the microbiome, and that this is associated with hair loss [7]. Cutibacterium spp. and Staphylococcus spp. account for about 90% of the microbiome in the healthy scalp, in addition to Corynebacterium spp., Streptococcus spp., Acinetobacter spp., and Prevotella spp. [8].
More background studies should be reviewed not only stating the taxonomy of scalp microbiome but also the physiology of scalp microbes
→ Thank you for your comment. This study focus on evaluating efficacy based on efficacy evaluation and distribution of microorganisms. Therefore, microbiome analysis before and after culture treatment focused on environmental analysis of scalp microbiome. We will reveal the physiology of scalp microbes through further research. I hope this response is suitable for your valuable indication
Line 122-123: Results should not describe M&M. Move it to M&M section
→ Thank you for your comment. As indicated, line 122-123 was moved to M&M section.
Same is the case in line 126 to 127. This should be mentioned in M&M
→ Thank you for your comment. As indicated, line 126-127 was moved to M&M section.
Line 145 to 151: Authors are just stating the M&Ms and not results
→ Thank you for your comment. As indicated, line 145-151 was moved to M&M section.
The whole of the results and discussion section looks like authors are re-writing the M&Ms. The whole results and discussion section should be re-written. The conclusion is just a repitition of results
→ Thank you for your comment. As indicated, whole parts in manuscript were revised as attached file.

Reviewer 2 Report
The authors present a protocol to illustrate the putative role of the microbiome in hair loss. I found the manuscript compelling and I do think it can be published. Nevertheless, I have the following minor comments:
Line 39: I'd suggest changing "Our skin..." to Human skin
Line 53: Change Propecia to Finasteride, to give consistency.
Line 55: Change 5a to the proper symbol.
Line 74: "a total of 28,000 reads" is vague, please develop further if possible.
Lines 84-85: Please rephrase
Figure 1: The resolution is low, also the use of subfigures makes it difficult to follow. I'd suggest dividing the figure or perhaps consider its placement as supplementary information.
Figure 6: See previous comment.
The last paragraph in the conclusions is confusing. Perhaps the wording can be improved to convey the authors' intention clearly.
Major comments:
The authors use the "data not shown" statement on several occasions. I'd suggest the inclusion of some of this data as supplementary material.
A major concern I have is the characterization of the Staphylococcus strain. The authors state this as novel. Yet provide little evidence or proof such is the case.
Plus, based on current data, this study adds support to the putative use of adenosine for hair loss. For more on this see:
10.1111/j.1346-8138.2008.00564.x
10.3390/molecules27072184
10.1046/j.0022-202x.2001.01570.x
https://pubmed.ncbi.nlm.nih.gov/24183218/
I think this should be part of the dissussion presented herein.
Author Response
Thank you for your kind letter, with regard to our manuscript together with comments. We are thankful to you for the very valuable suggestions through the whole manuscript. Thank you again for your kind considerations.
We tried to revise the manuscript as much as possible in line with suggestion made by the reviewer. I am herewith enclosing improved manuscript.
- Reviewer 2
The authors present a protocol to illustrate the putative role of the microbiome in hair loss. I found the manuscript compelling and I do think it can be published. Nevertheless, I have the following minor comments:
Line 39: I'd suggest changing "Our skin..." to Human skin
→ Thank you for your comment. As indicated, “Our skin” was revised as "Human skin” as below.
Human skin is home to millions of bacteria, fungi, and viruses that make up the skin microbiome Similar to intestinal microbes, skin microbes play an essential role in protecting against pathogen invasion, in the human immune system, and in decomposition of natural products [1–3].
Line 53: Change Propecia to Finasteride, to give consistency.
→ Thank you for your comment. As indicated, “Propecia” was revised as "Finasteride” as below.
The use of drug therapy with Minoxidil and Finasteride (MSD) is currently being attempted to treat, alleviate, or prevent hair loss.
Line 55: Change 5a to the proper symbol.
→ Thank you for your comment. As indicated, "5a” was revised as "5α” as below.
Minoxidil is assumed to induce hair growth through vasodilation and potassium channel opening, and Finasteride inhibits the activity of 5α-reductase, an enzyme acting on male hormone metabolism, to prevent male-type hair loss [5].
Line 74: "a total of 28,000 reads" is vague, please develop further if possible.
→ Thank you for your comment. As indicated, I correct the sentence line 74 as below
A total of 28,000 sequencing reads were obtained for 40 scalp samples for ordinary and hair loss patients, confirming secure data for analysis
Lines 84-85: Please rephrase
→ Thank you for your comment. As indicated, the sentence in lines 84-85 were rephrase as below.
As shown in Figure 1c, the six classes (Actinobacteria, Gammaproteobacteria, Clostridia, Betaproteobacteria, Bacilli, and Alphaproteobacteria) were showed dominant tendency at the class level (Figure 1c ).
Figure 1: The resolution is low, also the use of subfigures makes it difficult to follow. I'd suggest dividing the figure or perhaps consider its placement as supplementary information.
→ Thank you for your comment. As indicated, figure 1 is moved to supplementary materials as figure S1
Figure 6: See previous comment.
→ Thank you for your comment. As indicated, figure 6 is moved to supplementary materials as figure S2
The last paragraph in the conclusions is confusing. Perhaps the wording can be improved to convey the authors' intention clearly.
→ Thank you for your comment. As indicated, we correct last paragraph in the conclusions as revised manuscript
Major comments:
The authors use the "data not shown" statement on several occasions. I'd suggest the inclusion of some of this data as supplementary material.
→ Thank you for your comment. As indicated, we prepared the supplementary materials.
A major concern I have is the characterization of the Staphylococcus strain. The authors state this as novel. Yet provide little evidence or proof such is the case.
→ Thank you for your comment. As a result of microbiome analysis, Staphylococcus accounted for 90% of the strains of the scalp. The final strain was selected by performing an active evaluation on the strains secured in this study, and it was confirmed that it was a new strain through DNA sequencing. Therefore, we judged that the strain was novel. I hope this response is suitable for your valuable indication
I hope the improved version will be acceptable for publication in Molecules.
Yours sincerely,
Prof. Se Chan Kang

Round 2
Reviewer 2 Report
The manuscript has been improved significantly. Still, there are some minor issues. For instance there are some typographic or style error; e.g. Sicaria or Shamppo.
Thus I suggest an extensive revision.
On the other hand, the added figures (3&4) have low resolution. I think the axes should be resized. Some of the legends are also difficult to read. Also, I think that Supplementary Figure 1 could be split in two, to improve the resolution.
Finally, while commenting on the clinical trials it states cosmetic guidelines were used, but no indications on which exactly.
Author Response
Molecules
Manuscript number: molecules-1736817
Title: Staphylococcus epidermidis Cicaria, a novel strain derived from the human microbiome, and its efficacy as a treatment for hair loss
Dear, Editor in Chief and reviewer 2
Thank you for your kind letter, with regard to our manuscript together with comments. We are thankful to you for the very valuable suggestions through the whole manuscript. Thank you again for your kind considerations.
We tried to revise the manuscript as much as possible in line with suggestion made by the reviewer 2. I am herewith enclosing improved manuscript.
- Reviewer 2
The manuscript has been improved significantly. Still, there are some minor issues. For instance there are some typographic or style error; e.g. Sicaria or Shamppo. Thus I suggest an extensive revision.
→ Thank you for valuable indication. As indicated, I reviewed the manuscript carefully and corrected the typographic or style error as attached one.
On the other hand, the added figures (3&4) have low resolution. I think the axes should be resized. Some of the legends are also difficult to read. Also, I think that Supplementary Figure 1 could be split in two, to improve the resolution.
→ Thank you for your comment. As indicated, the figures which you mentioned were divided and replaced, respectively as attached one.
Finally, while commenting on the clinical trials it states cosmetic guidelines were used, but no indications on which exactly.
→ Thank you for your comment. In this study, the clinical study focused on the distribution of microbiomes, and we could not confirm the part you mentioned. We will do that through a follow-up study. I hope this response is suitable for your valuable indication.
I hope the improved version will be acceptable for publication in Molecules.
Yours sincerely,
Prof. Se Chan Kang
